# DREAMMAKEUP: FACE MAKEUP CUSTOMIZATION USING LATENT DIFFUSION MODELS

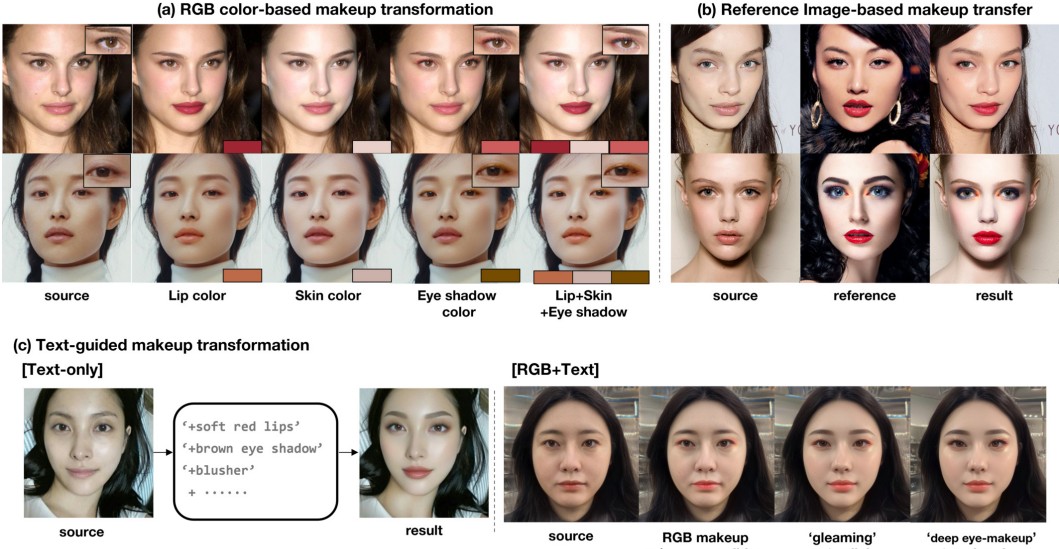

Figure 1: **Representative usage of DreamMakeup**. (a) Makeup transformation with RGB target color. (b) Makeup transfer with reference image. (c) Text-guided makeup.

## ABSTRACT

The exponential growth of the global makeup market has paralleled advancements in virtual makeup simulation technology. Despite the progress led by GANs, their application still encounters significant challenges, including training instability and limited customization capabilities. Addressing these challenges, this paper introduces *DreamMakup*: a novel Diffusion model based Makeup Customization, leveraging the inherent advantages of diffusion models for superior controllability and precise real-image editing. DreamMakeup employs early-stopped DDIM inversion to preserve the facial structure and identity while enabling extensive customization through various conditioning inputs such as reference images, specific RGB colors, and textual descriptions. Our model demonstrates notable improvements over existing GAN-based frameworks, improved customization, color-matching capabilities, and compatibility with textual descriptions or LLMs with affordable computational costs. Project page is available at here.

## 1 INTRODUCTION

The global makeup market size is valued at billions of dollars, and virtual makeup simulation technology is considered to be a rapidly growing sector within the beauty industry. This is driven by the increasing adoption of AI or AR technologies for virtual try-on experiences by beauty brands and consumers alike. Besides its industrial importance, face makeup customization is also an interesting problem in terms of generative modeling and editing. Specifically, one may have to disentangle

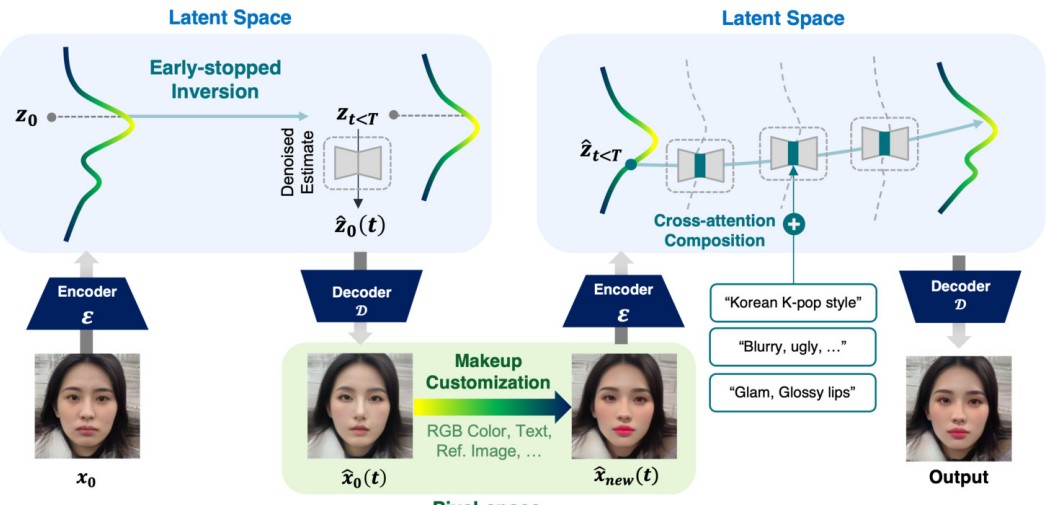

Figure 2: **Overview of *DreamMakeup* pipeline**. The key principle of our framework is to apply fine-grained guidance in high-dimensional pixel-domain during reverse sampling. After local makeup customization in pixel space, text prompts are leveraged to harmonize such local variations with a consistent global style in cross-attention space.

and stylize each facial attirbute in their independent style, while its composition should be well harmonized with a consistent aesthetic style.

So far, virtual face makeup modeling is mainly driven by generative adversarial networks (GANs) (Li et al. (2018); Jiang et al. (2019); Deng et al. (2021); Liu et al. (2023); Yang et al. (2022)). Despite its advancements, GAN-based frameworks face inherent instability in adversarial training, leading to several limitations. For instance, GANs in real-world applications demands extensive collections of non-makeup and makeup facial images with diverse poses, expressions, styles, and complex backgrounds. Furthermore, existing GAN-based methods are not fully customizable and lack controllability. Specifically, most of these frameworks only support makeup transfer tasks, inherently requiring reference target images. In many business contexts, users may seek to simulate facial makeup with a more degree of freedom, e.g. test with specific RGB colors of new cosmetic products, or linguistic descriptions such as ``Glam makeup style with glossy lips".

In response to these challenges, this paper explores the adoption of diffusion models, recognized for their superior controllability and real-image editing capabilities. Diffusion models offer several advantages for facial beauty simulation. For instance, we can control the reverse sampling process using the enriched text-conditions. Moreover, it supports various style customization using LoRAs (Hu et al. (2021)) supported by the vibrant user community. By employing techniques such as DDIM inversion (Song et al. (2020); Mokady et al. (2023)), diffusion models well preserve the overall structure and subject identity of given facial images, while retaining rich editing capabilities.

To this end, we introduce *DreamMakeup*, a novel diffusion-based makeup customization distinguished by its advanced sampling guidance. This model is fully compatible with a variety of conditionings to steer the makeup process, ranging from reference images and specific RGB colors to textual descriptions of desired makeup looks. As shown in Fig. 2, given pre-trained latent diffusion models (LDM), we commence by inverting facial images $x$ into latents $z_t$ through early-stopped DDIM inversion. Subsequently, we approximate the denoised estimate $\mathbb{E}[z_0 \mid z_t]$ and decode it back into the pixel space, preserving the facial structure attributed to the inversion process. Then, we stylize these facial representations in pixel space through transformations such as histogram matching, RGB color matching, or warping, toward a targeted makeup style. Resuming the reverse sampling process from these transformed representations with proper makeup prompts yields harmonized facial makeup outcomes. Our contributions are summarized as follows:

- We introduce *DreamMakeup*, a novel diffusion-based human face makeup framework that customizes facial makeup with advanced pixel-space sampling guidance. To our knowl-

edge, this is the first to offer a fully customizable facial makeup application catering to a wide range of user preferences including text descriptions, colors, and reference images.

- DreamMakeup is computationally affordable as it does not fundamentally require fine-tuning. Moreover, we early-stop DDIM inversion process to preserve the facial structures which further saves the computational costs. This enables fast overall inference ($< 4$ seconds for color transfer w/ SD v1.5 (Rombach et al., 2022), NVIDIA GeForce RTX 4090).
- DreamMakeup outperforms real-world global AI makeup services in color maekup task, and state-of-the-art GAN-based frameworks in makeup transfer tasks, setting a new benchmark for facial makeup simulation. Furthermore, we demonstrate that our framework can be easily integrated with other foundational models, including Large Language Models (LLMs) and classifier of facial structure, broadening the horizon for virtual makeup.

## 2 RELATED WORKS

Makeup transfer aims to modify a facial image to reflect a chosen makeup style, with numerous approaches developed using Generative Adversarial Networks (GANs). Notably, BeautyGAN (Li et al. (2018)) employed histogram matching to preserve color from the reference image. LADN (Gu et al. (2019)) used local discriminators for heavy makeup. PSGAN (Jiang et al. (2019)) enhanced style controllability through matrices and addressed pose misalignments with attention mechanisms. SCGAN (Deng et al. (2021)) tackled pose issues with style codes. RamGAN (Xiang et al. (2022)) improved makeup transfer with regional attention, and EleGANt (Yang et al. (2022)) offered flexible control over arbitrary regions using attention mechanisms, reducing computational demands.

However, GAN-based methods typically require large datasets of makeup and no-makeup images for training and reference images for inference, limiting application diversity and customizability. DreamMakeup overcomes these limitations by utilizing foundational diffusion models and open-source LoRAs for image generation, enabling replication of makeup styles from references or modification via RGB values or textual prompts. This approach enhances controllability and versatility without relying on extensive datasets.

## 3 PRELIMINARY

Diffusion models aim to generate samples from the Gaussian noise through iterative denoising processes. Since pixel-space diffusion models are computationally heavy, the latent diffusion model (LDM) (Rombach et al. (2022)) operates the diffusion process on latent space instead of pixel space. Given a pixel-space clean sample $\boldsymbol{x} \sim p_{\text{data}}(\boldsymbol{x})$, Rombach et al. (2022) leverages an autoencoder

$$\mathcal{E} : \mathbb{R}^d \to \mathbb{R}^k, \mathcal{D} : \mathbb{R}^k \to \mathbb{R}^d, \boldsymbol{x} \simeq \mathcal{D}(\mathcal{E}(\boldsymbol{x})), \forall \boldsymbol{x} \sim p_{\text{data}}(\boldsymbol{x}), \tag{1}$$

where $\mathcal{E}$ is the encoder, $\mathcal{D}$ is the decoder, and the dimension of latent space $k < d$. After training $\mathcal{E}$ and $\mathcal{D}$, one can define the forward and reverse diffusion process within the latent space $\boldsymbol{z} = \mathcal{E}(\boldsymbol{x})$.

The forward process is defined as a Markov chain, characterized by forward conditional densities:

$$\begin{aligned} p(\boldsymbol{z}_t \mid \boldsymbol{z}_{t-1}) &= \mathcal{N}(\boldsymbol{z}_t \mid \beta_t \boldsymbol{z}_{t-1}, (1 - \beta_t)I) \\ p_t(\boldsymbol{z}_t \mid \boldsymbol{z}_0) &= \mathcal{N}(\boldsymbol{z}_t \mid \sqrt{\bar{\alpha}}\boldsymbol{z}_0, (1 - \bar{\alpha})I), \end{aligned} \tag{2}$$

with $\boldsymbol{z}_t \in \mathbb{R}^k$ representing the noisy latent variable at a timestep $t \leqslant T$ that has the same dimension as $\boldsymbol{z}_0 = \mathcal{E}(\boldsymbol{x}_0)$ for $\boldsymbol{x}_0 \sim p_{\text{data}}(\boldsymbol{x})$, and $\beta_t$ denotes an increasing sequence of noise schedule where $\alpha_t := 1 - \beta_t$ and $\bar{\alpha}_t := \Pi_{i=1}^t \alpha_i$. The goal of training LDM is to obtain a residual denoiser $\boldsymbol{\epsilon}_{\theta*}$:

$$\theta^* = \arg\min_\theta \mathbb{E}_{\mathcal{E}(\boldsymbol{x}_0), t, \boldsymbol{\epsilon} \sim \mathcal{N}(0, I)} \big[ \|\boldsymbol{\epsilon}_\theta(\boldsymbol{z}_t, t) - \boldsymbol{\epsilon}\| \big]. \tag{3}$$

The reverse sampling from $q(\boldsymbol{z}_{t-1} | \boldsymbol{z}_t, \boldsymbol{\epsilon}_{\theta*}(\boldsymbol{z}_t, t))$ is then achieved by

$$\boldsymbol{z}_{t-1} = \frac{1}{\sqrt{\alpha_t}} \Big( \boldsymbol{z}_t - \frac{1 - \alpha_t}{\sqrt{1 - \bar{\alpha}_t}} \boldsymbol{\epsilon}_{\theta*}(\boldsymbol{z}_t, t) \Big) + \tilde{\beta}_t \boldsymbol{\epsilon}, \tag{4}$$

where $\boldsymbol{\epsilon} \sim \mathcal{N}(0, \boldsymbol{I})$ and $\tilde{\beta}_t := \frac{1 - \bar{\alpha}_{t-1}}{1 - \bar{\alpha}_t} \beta_t$. We will omit $*$ in $\theta^*$ in the rest of the paper. After reverse sampling, the generated latent $\tilde{\boldsymbol{z}}_0$ is decoded to the pixel space as $\tilde{\boldsymbol{x}}_0 = \mathcal{D}(\tilde{\boldsymbol{z}}_0)$.

To accelerate sampling, DDIM (Song et al. (2020)) proposes an alternative sampling method:

$$\boldsymbol{z}_{t-1} = \sqrt{\bar{\alpha}_{t-1}}\hat{\boldsymbol{z}}_0(t) + \sqrt{1 - \bar{\alpha}_{t-1} - \eta^2\tilde{\beta}_t^2}\boldsymbol{\epsilon}_\theta(\boldsymbol{z}_t, t) + \eta\tilde{\beta}_t\boldsymbol{\epsilon}, \tag{5}$$

where $\eta \in [0, 1]$ is a stochasticity parameter, and $\hat{\boldsymbol{z}}_0(t)$ is the denoised estimate which can be equivalently derived using Tweedie's formula (Efron (2011)):

$$\hat{\boldsymbol{z}}_0(t) := \frac{1}{\sqrt{\bar{\alpha}_t}}(\boldsymbol{z}_t - \sqrt{1 - \bar{\alpha}_t}\boldsymbol{\epsilon}_\theta(\boldsymbol{z}_t, t)). \tag{6}$$

For text-guided sampling, we train the diffusion model with textual embedding $c$. We will often omit $c$ from $\boldsymbol{\epsilon}_\theta(\boldsymbol{x}_t, t, c)$ to avoid notational complexity.

## 4 DIFFUSION-BASED MAKEUP CUSTOMIZATION

Given an input non-makeup human facial image $\boldsymbol{x}_0$, our main goal is to (**a**) customize the makeup style with coarse (e.g. RGB color) to fine (e.g. reference makeup image) level information, while (**b**) preserving the overall facial structure and subject identity.

To achieve this, *DreamMakeup* leverages various user preferences for conditional guidance, e.g. target color, reference image, and textual make-up description. As shown in Fig. 2, the customization process consists of three main phases: (**1**) early-stopped DDIM inversion to impose structural consistency, (**2**) pixel-space optimization to guide the sampling process towards target makeup style, and (**3**) reverse sampling with cross attention composition to accommodate various textual makeup descriptions simultaneously.

One of our primary contributions is to integrate a pixel-space makeup customization during reverse sampling process given the decoded intermediate estimates $\hat{\boldsymbol{x}}_0(t) = \mathcal{D}(\hat{\boldsymbol{z}}_0(t))$. While makeup customization demands delicate control over low-level visual appearance, such features like color and edges are often encoded nonlinearly in low-dimensional latents, making direct customization challenging in latent space. To address this, we hijack the latents to the pixel-domain, where low-level visual features can be fully identified and guided. This *one-step* fine-grained customization in the *pixel-domain* is a significant departure from existing *multi-step* latent/attention-space guidance (Hertz et al., 2023; Kwon & Ye, 2022; Mokady et al., 2023). Moreover, we empirically demonstrate that DreamMakeup can be integrated with Large Language Models (LLM) or facial classifiers, paving a new path for virtual makeup pipeline design. Additional experimental details and pseudo-code are available in the appendix.

### 4.1 EARLY-STOPPED INVERSION

To maintain the identity of the original input face during the sampling process, we first leverage DDIM inversion which is an iterative reverse simulation of the ODE flow in the limit of small steps. By setting $\eta = 0$ in Eq. (5), the DDIM inversion (Mokady et al. (2023); Song et al. (2020)) is defined as follows:

$$\boldsymbol{z}_t = \frac{\sqrt{\bar{\alpha}_t}}{\sqrt{\bar{\alpha}_{t-1}}}\boldsymbol{z}_{t-1} - \sqrt{\bar{\alpha}_t}\left(\sqrt{\frac{1}{\bar{\alpha}_{t-1}} - 1} - \sqrt{\frac{1}{\bar{\alpha}_t} - 1}\right)\boldsymbol{\epsilon}_\theta(\boldsymbol{z}_{t-1}, t, c). \tag{7}$$

Note that we fix the textual condition $c$, e.g. `"a photography of a woman"`. While the conventional process inverts $\boldsymbol{z}_0$ to $\boldsymbol{z}_T$, we terminate the inversion at $t^* \leqslant T$ to reduce the computational burdens and ensure structural consistency. Specifically, Figure 3 shows that the denoised estimate $\hat{\boldsymbol{z}}_0(t^*)$ from early-stopped $\boldsymbol{z}_{t*}$ decodes faithfully to the original sample $\boldsymbol{x}_0$. This allows us to directly guide $\hat{\boldsymbol{x}}_0(t^*) = \mathcal{D}(\hat{\boldsymbol{z}}_0(t^*))$ in a pixel space to enforce the target makeup style. While we early-stop DDIM inversion at $t^* < T$, most of inversion-based frameworks (Tumanyan et al., 2023; Parmar et al., 2023; Park et al., 2024) typically terminate at $t^* = T = 1000$ which slows down inference.

### 4.2 PIXEL-SPACE MAKEUP CUSTOMIZATION

Our next goal is to transform $\hat{\boldsymbol{x}}_0(t^*)$ in a manner that accurately emulates the desired makeup appearance indicated by reference image or a target RGB color. To this end, we introduce a pixel-space transformation $\mathcal{T}(\cdot, \cdot) : \mathbb{R}^{H \times W \times 3} \times X \to \mathbb{R}^{H \times W \times 3}$, offering multiple variants of $\mathcal{T}$ to enable

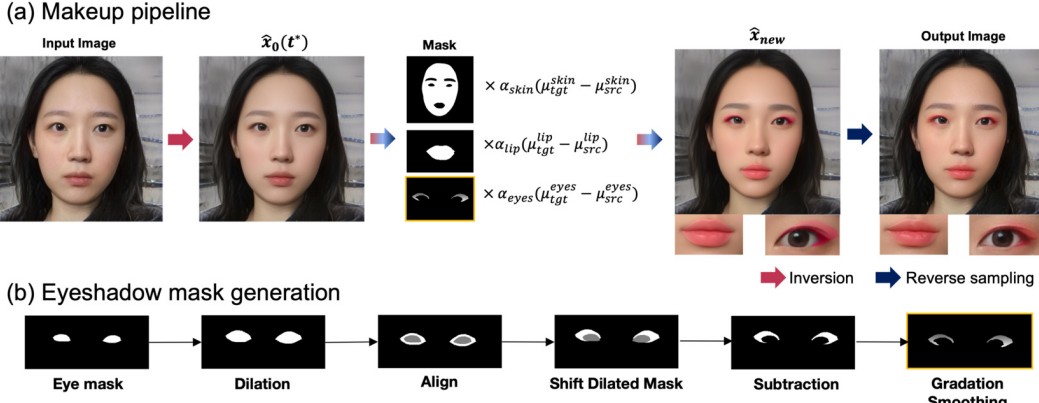

Figure 3: **RGB color-based makeup transformation.** The mean RGB values within the masked area are adjusted with a scale $\alpha$ to match the target RGB values. The final output image is generated by reverse sampling from $\hat{x}_{new}$. We manipulate the eye mask to reproduce eyeshadow mask.

fine-grained makeup customization. Here $X$ varies for different references, e.g. target RGB color, reference image, etc.

### 4.2.1 MAKEUP TRANSFORMATION WITH RGB COLOR

We first delineate an intuitive color transfer function that imposes the color characteristics of the reference makeup palette color on the source image. Let $\mu_{src}(\hat{x}_0(t^*))$, $\sigma_{src}(\hat{x}_0(t^*))$ represents the RGB mean and standard deviation of $\hat{x}_0(t^*)$ computed across spatial dimensions. Given a reference color $\mu_{tgt}$ and respective standard deviation $\sigma_{tgt}$, the color transfer function $\mathcal{T}_{RGB}$ is defined as

$$\hat{x}_{new}(t^*) = \mathcal{T}_{RGB}\big(\mu_{src}(\hat{x}_0(t^*)), \mu_{tgt}; \alpha\big) = \frac{\sigma_{src}(\hat{x}_0(t^*))}{\sigma_{tgt}}\Big(\hat{x}_0(t^*) - \alpha\big[\mu_{src}(\hat{x}_0(t^*)) - \mu_{tgt}\big]\Big),$$
(8)

where $0 \leqslant \alpha \leqslant 1$ represents a transfer scale. For simplicity, we empirically set $\sigma_{src}(\hat{x}_0(t^*)) = \sigma_{tgt}$.

**Color makeup composition.** The proposed color transfer supports various independent attribute compositions. Specifically, we may eager to transfer different $\mu_{tgt}$ for each lips, eye shadow, skin foundation, etc. For this, we segment each interested facial attribute using a segmentation model (Yu et al., 2018) pre-trained in a pixel domain. We observed that the inverted image $\hat{x}_0(t^*)$ is well segmented by the pre-trained model, owing to its high similarity with the original image $x_0$.

While lips and facial masks are readily available via Yu et al. (2018), eyeshadow masks require additional processing. For this, we reconstruct the eyeshadow mask by manipulating the eye mask through additional transformation such as dilation and shifting (Figure 3). To achieve a more seamless and natural makeup integration, we smooth the edges of the eyeshadow mask. Specifically, the binary eyeshadow mask is multiplied by a gradient mask, with its weights progressively increasing from the inner to the outer edge. This allows fine control, enabling users to adjust the gradient decay rate for a more natural appearance. Any artifacts from this color transfer process are further refined through reverse diffusion sampling (More details in Section 4.3).

### 4.2.2 MAKEUP TRANSFER WITH REFERENCE IMAGE

The transformation $\mathcal{T}$ can be varied depends on the downstream task. In context of conventional makeup transfer tasks (Li et al., 2018), we simulate the makeup style of reference image through warping and histogram matching transformations.

First, histogram matching aligns the color distributions of the lip, eye shadow, and skin with reference. Then, the eyes of the source face are aligned with the reference through a series of warping transformations, including segmentation, dilation, affine, and diffeomorphic transformations, to ensure precise registration. This ensures that every pixel within the dilated mask area of the reference image corresponds to the appropriate region on the source image. The eyeshadow mask is then

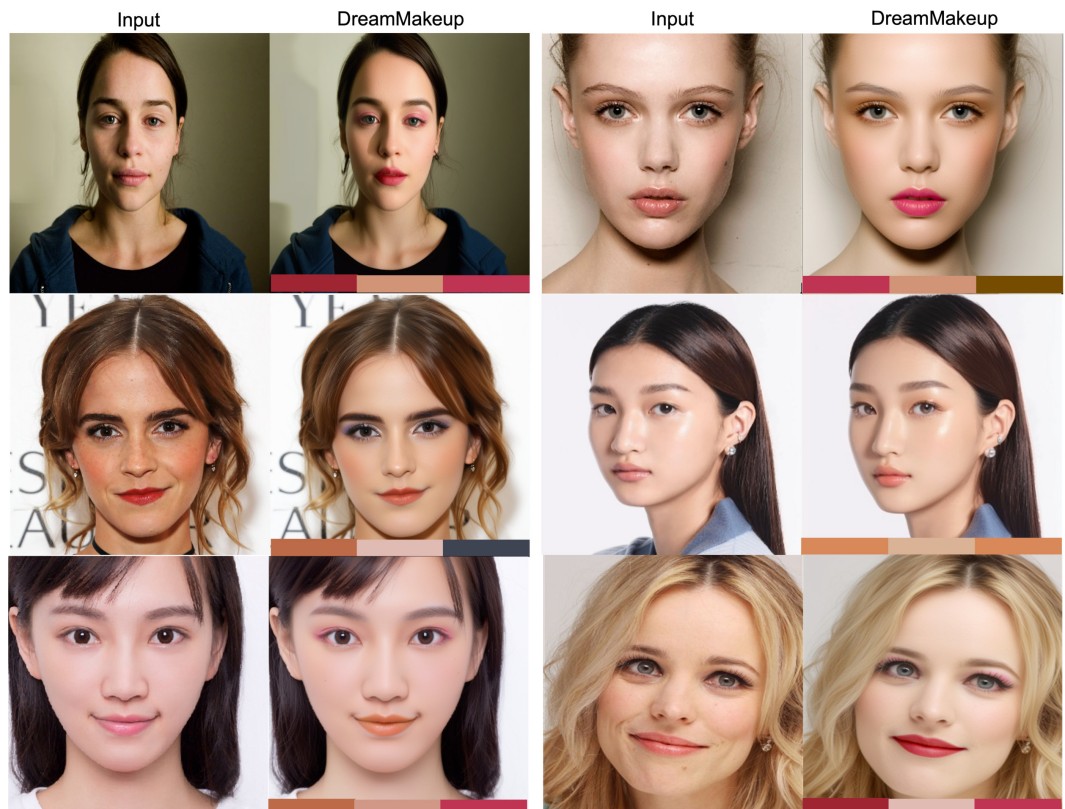

Figure 4: The color-based makeup examples by DreamMakeup with SDXL. Each color chip represents the color of lips, skin and eyeshadow.

transferred from the reference to the source with additional smoothing for natural edge transitions. These steps enable the seamless adoption of tones and styles of the reference image, ensuring a natural and accurate makeup transfer.

### 4.3 MAKEUP HARMONIZATION IN CROSS-ATTENTION LAYERS

After pixel-space guidance, the output $\hat{x}_{new}$ preserves facial structural consistency while coarsely following the desired makeup style in the local facial attributes, e.g. lips, eye shadow, etc. Our next goal is to stylize $\hat{x}_{new}$ with text guidance to harmonize such local variations with a fine-grained consistent global aesthetic style, e.g. "Korean K-Pop style", "Nude makeup", etc. Moreover, since the local transfer is based on coarse masks, it contains some unnatural discontinuity ($\hat{x}_{new}$ in Figure 3). Thus, the following refinement process aims to: (**a**) align $\hat{x}_{new}$ with specified textual descriptions and (**b**) enhance overall image quality.

Note that it is challenging to reflect the compositions of detailed linguistic conditions while preserving subject identity with conventional diffusion-based frameworks (Hertz et al., 2022; Tumanyan et al., 2023; Brooks et al., 2023). To address this, we refine the image by integrating semantic makeup features directly into the cross-attention layer, where the overall spatial structure is maintained. Specifically, let $\boldsymbol{Q}_{t,l} \in \mathbb{R}^{P_l^2 \times d_l}$ represent the spatial query in $l$-th cross attention layer of U-Net with resolution $P_l$ and dimension $d_l$ at time $t$. We will often omit $t$ and $l$ for notational simplicity. Given context vectors $\boldsymbol{C} \in \mathbb{R}^{N \times d_c}$, let $\boldsymbol{K}, \boldsymbol{V} \in \mathbb{R}^{d_c \times d_l}$ denote key and value matrices, respectively, where $N$ refers to number of tokens, $\boldsymbol{K} = \boldsymbol{C}\boldsymbol{W}_{K,l}$ and $\boldsymbol{V} = \boldsymbol{C}\boldsymbol{W}_{V,l}$ with linear maps $\boldsymbol{W}_{K,l}, \boldsymbol{W}_{V,l} \in \mathbb{R}^{d_c \times d_l}$. Then, let $\boldsymbol{K}_s$ represents a $s$-th makeup concept key $\boldsymbol{K}_s$, where $\boldsymbol{K}_{main}$ comes from the prompt used in inversion, i.e. "a photography of a woman". Define $\boldsymbol{V}_s, \boldsymbol{V}_{main}$ similarly. Then, we update the spatial query as follows:

$$\boldsymbol{Q}^{new} = \mathrm{softmax}\Big(\frac{\boldsymbol{Q}\boldsymbol{K}_{main}^T}{\sqrt{d}}\Big)\boldsymbol{V}_{main} + \frac{1}{M}\sum_{s=1}^{M}\alpha_s \mathrm{softmax}\Big(\frac{\boldsymbol{Q}\boldsymbol{K}_s^T}{\sqrt{d}}\Big)\boldsymbol{V}_s. \tag{9}$$

| Reference | Source | PSGAN | SCGAN | EleGANt | Ours |
|---|---|---|---|---|---|

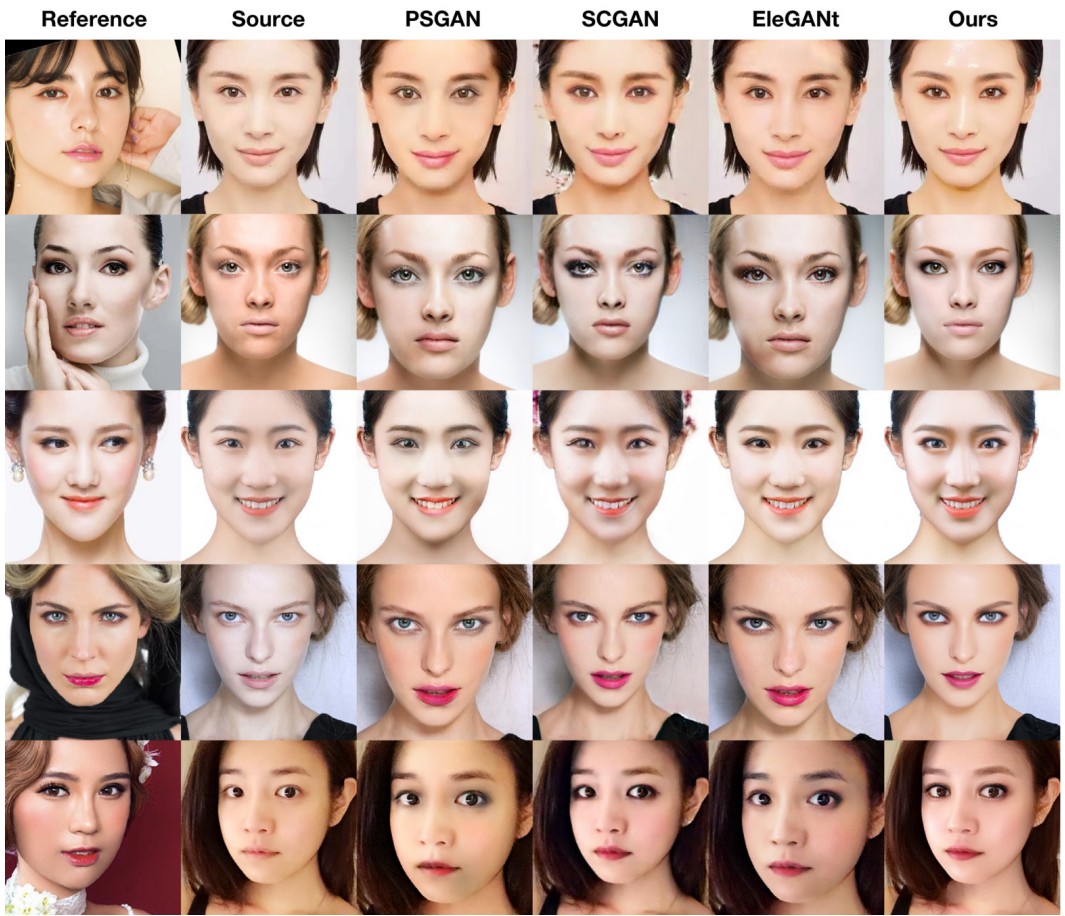

Figure 5: The makeup transfer by DreamMakeup from reference images with various poses and makeup.

Note that the degree and direction of each $s$-th makeup concept can be controlled individually with $\alpha_s$, where $\alpha_s < 0$ for negative makeup prompts, e.g. `ugly`, `blurry`, `low-res`, etc. This linear combination incorporates detailed makeup descriptions independently, while preserving the main ODE sampling path derived from inversion. We empirically observed that the proposed sampling method works well even without external regional masks, which are commonly used in diffusion-based customization frameworks (Gu et al., 2024; Kwon et al., 2024). This may be attributed to the global nature of makeup prompts focusing on aesthetic style, and the distinct separation of facial attributes within semantic features. Overall pipeline is summarized in Figure 2 and pseudo-code in appendix.

## 5 EXPERIMENTAL RESULTS

### 5.1 EXPERIMENT SETTINGS

During inference, we used the Makeup Transfer (MT) dataset (Li et al. (2018)) for both source and reference images. Also, synthetic face dataset is used. Additionally, we employed artificially generated images for Asian women as a non-makeup source image. We utilized Stable Diffusion (SD) v1.5 and SDXL (Podell et al., 2023) as our base model, and further leveraged additional public LoRA weights and pre-trained models released in CivitAI, an open-source generative AI community. Specifically, Dreamshaper[1], ArienMixXL [2], and BKG1[3] LoRA weights are mainly used. For facial segmentation, we utilized BiSeNet (Yu et al. (2018)). For cross attention composition, we set the

---

[1]https://civitai.com/models/4384/dreamshaper

[2]https://civitai.com/models/118913/sdxl-10-arienmixxl-asian-portrait

[3]https://civitai.com/models/203947/beautiful-korean-girl-bkgv1

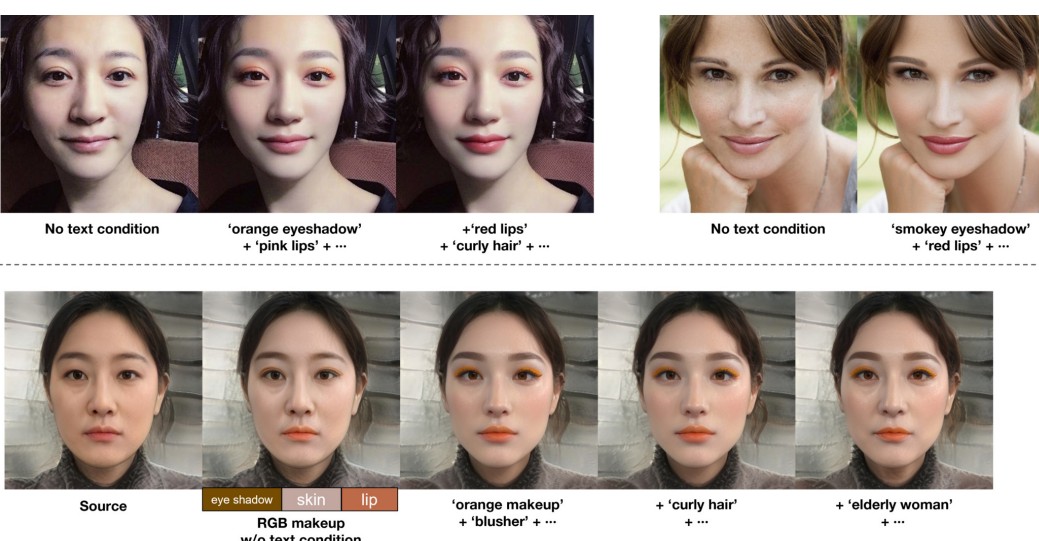

Figure 6: The virtual skin, lip, eye shadow makeup, and their combination by DreamMakeup. The left images are generated using SD 1.5, and the right images are made using SDXL.

Figure 7: **Text-guided makeup transformation**. Top row: text-guided makeup transformation. Bottom row: text + RGB color guided transformations.

scaling factor for each makeup concept $\alpha_s$ ranging from 0.1 to 0.7. For comparison, we tested three state-of-the-art GAN-based makeup transfer methods, PSGAN (Jiang et al. (2019)), SCGAN (Deng et al. (2021)), and EleGANt (Yang et al. (2022)). More experimental details are in appendix.

| Method | LPIPS ↓ | CLIP-I ↑ | Makeup Artists | | Non Artists | |
|--------|---------|----------|--------|---------|--------|---------|
| | | | Detail ↑ | Quality ↑ | Detail ↑ | Quality ↑ |
| PSGAN | 0.1879 | 0.7421 | 1.49 | 1.74 | 2.64 | 2.82 |
| SCGAN | 0.0819 | 0.7253 | 2.00 | 2.11 | 3.03 | 2.95 |
| EleGANt | 0.1877 | 0.7662 | 3.04 | 3.12 | 3.67 | 3.72 |
| Ours | **0.0667** | **0.7694** | **3.42** | **3.42** | **3.95** | **4.00** |

(a) Quantitative results on makeup transfer tasks.

| Method | Beauty score | Makeup Artists | | Non Artists | |
|--------|--------------|--------|---------|--------|---------|
| | | Detail ↑ | Quality ↑ | Detail ↑ | Quality ↑ |
| Service A | 2.90 | 4.08 | 1.97 | 3.69 | 2.77 |
| Service B | 3.27 | 3.75 | 2.61 | 3.14 | 2.83 |
| Ours | **3.38** | **4.22** | **4.19** | **3.93** | **4.27** |

(b) Comparisons with global AI makeup services.

Table 1: Quantitative comparisons on makeup transfer task and color-based makeup transformation.

## 5.2 RESULTS

### 5.2.1 QUALITATIVE RESULTS

Figure 6 illustrates the RGB color-based makeup transformations applied to both synthetic and natural images. First three rows reflect the application of the eye shadow, skin, and lip colors indicated in

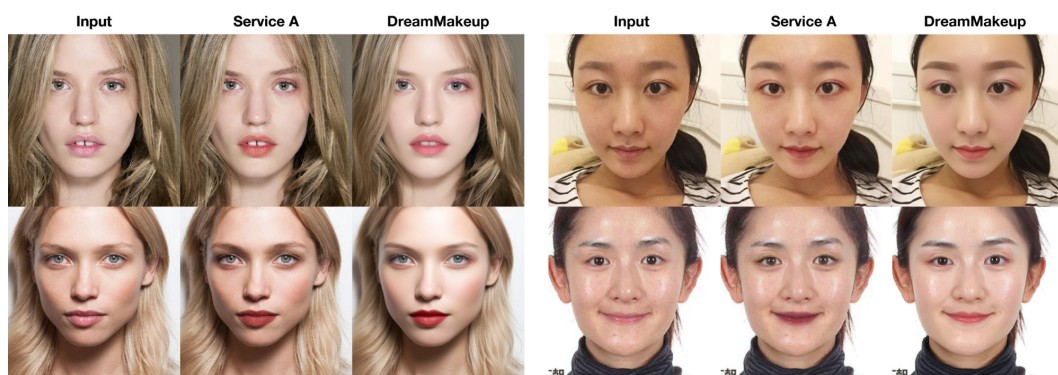

Figure 8: Comparisons of DreamMakeup with Other Global Mobile AI Makeup Services

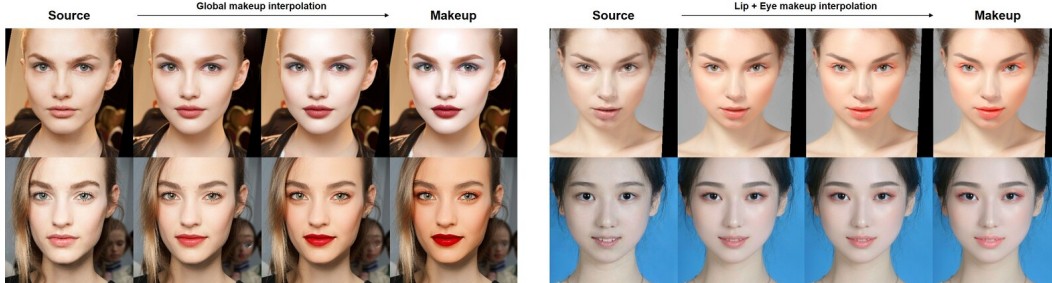

Figure 9: Results on global and local makeup interpolation

the bottom right corner. Bottom row presents the combined results for each column's corresponding colors. DreamMakeup effectively applies the specified RGB colors to the respective facial regions.

Figure 5 compares the results with existing methods (PSGAN, SCGAN, and EleGANt). Our method outperforms the others, with PSGAN and SCGAN displaying artifacts like identity loss and color bleeding. While EleGANt generally performs well, it exhibits issues such as dark artifacts in the hand-covered region (row 2), weaker eye makeup or changed spatial layout. In contrast, our approach produces cleaner, artifact-free results, accurately replicating the reference makeup.

### 5.2.2 QUANTITATIVE COMPARISON

We further evaluate DreamMakeup against other baselines using quantitative metrics for both color makeup and makeup transfer tasks. For the makeup transfer task, we assess LPIPS and CLIP image similarity to measure identity preservation. Additionally, we gathered user feedback from 10 expert makeup artists and 24 non-expert participants, asking them to rank 10 randomly selected output images generated by different models. The evaluation criteria focused on two key aspects: how closely the makeup style in the output matched the target image (visual details) and the overall makeup quality. Ratings were provided on a scale of 1 to 5. As shown in Table 1a, DreamMakeup consistently outperformed other models.

For the color-based makeup task, we conducted comprehensive comparative studies with two global AI makeup services, which are featured in mobile beauty applications with over 50M downloads. To avoid potential conflicts and licensing issues, we do not disclose the specific service names. Despite their widespread use, these services exhibit limitations in customization, such as restricted color presets. We collected 100 images from each service, applying randomized makeup styles, and assessed the beauty score (Xu et al., 2019). Additionally, we conducted a more extensive user preference study using 300 images, evaluated by 10 makeup artists and 24 non-expert participants. As summarized in Table 1b, DreamMakeup significantly outperformed the other methods.

### 5.2.3 TEXT-GUIDED MAKEUP TRANSFORMATION

To demonstrate the efficacy of text guidance, we apply makeup on the source image with (1) only using text guidance with cross attention composition, and (2) using text guidance along with RGB color transfer. Figure 7 illustrates that text guidance effectively facilitates makeup transformation,

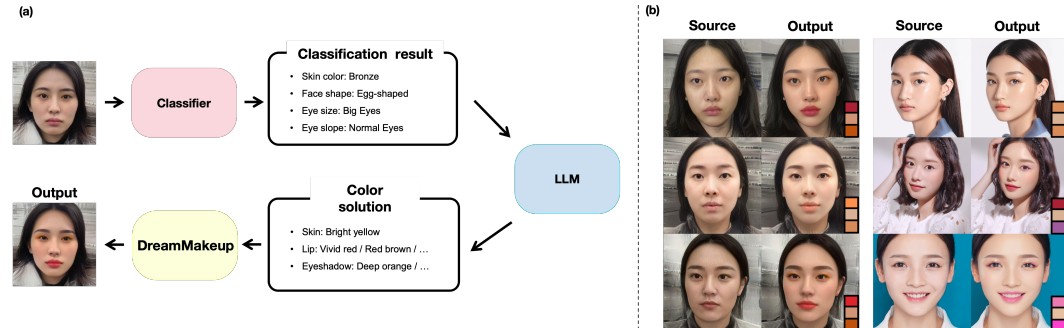

Figure 10: (a) DreamMakeup with integration with a classifier and an LLM. (b) From the source image, we can apply makeup based on the solution provided by the classifier and LLM. The color chips represent the color of lips, skin, and eye shadow from top to bottom.

and concisely stylizes the RGB-based makeup transformations. Specifically, the color transfer in the second column establishes the overall color distribution, and linguistic conditioning further improves the natural appearance.

### 5.2.4 MAKEUP INTERPOLATION

By adjusting a transfer scale $\alpha$ in equation 8, we can control the makeup intensity. As $\alpha$ converges towards 0, the result increasingly resembles the source image. The intensity of the makeup can be controlled independently for different makeup regions, such as lips and eyes (Figure 9).

### 5.3 INTEGRATION WITH LLM

DreamMakeup can demonstrate improved performance by integrating with Large Language Models (LLMs) for personalized makeup recommendations. By harnessing the exceptional inference capabilities of LLMs, DreamMakeup selects makeup colors that are harmonious with the characteristics of the source image. We provide a pipeline in Fig. 10. This pipeline involves an initial extraction of facial attributes such as skin tone and facial structure from the source image via a classifier. This information is then conveyed to the LLM, which determines the most appropriate makeup colors for various facial regions, including the skin, eyes, and lips. DreamMakeup subsequently utilizes these recommendations to generate the final makeup-enhanced image.

These components (classifier, LLM, and DreamMakeup) operate sequentially during inference but are trained independently. The classifier was trained using ResNet50 on a dataset of 1,000 artificially generated images of Asian women, annotated with facial information. The LLM is trained on the dolly-v2-3b model with a specialized QnA dataset from beauty professionals. As illustrated in Figure 10(b), the LLM adeptly matches skin, eye shadow, and lip colors to the source image, facilitating the application of these colors by DreamMakeup. The process ensures that the selected colors are well-suited to the source image, leading to an effectively applied makeup look. This demonstrates the potential of integrating classifiers and LLMs in DreamMakeup to provide customized makeup solutions based on in-depth analysis of facial features, thereby enhancing the personalization and effectiveness of makeup applications.

## 6 CONCLUSION

In this paper, we introduced DreamMakeup, a novel approach to customizing facial makeup utilizing the diffusion model. Our method leverages RGB colors or textual descriptions with pixel-space sampling guidance, ensuring precise makeup customization. Additionally, we demonstrated DreamMakeup's effectiveness in makeup transfer tasks through the application of a facial structure classifier and LLM. Notably, our approach is computationally efficient, paving the way for practical makeup customization solutions.

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

## A  PSEUDO CODE

We provide the pseudo-code of DreamMakeup for RGB and textual guidance in Algorithm 1. Makeup transfer based on a reference image can be easily implemented using the same method, substituting the transformation $\mathcal{T}_{RGB}$ with $\mathcal{T}_{ref}$ and employing warping and histogram matching algorithms instead of RGB matching.

---

**Algorithm 1** DreamMakeup with RGB and text guidance

---

**Input:** Source image $x_0$, early-stop timestep $t^* \leqslant T$, RGB scaling coefficient $0 \leqslant \alpha \leqslant 1$, target RGB color $\mu_{tgt}$, reference makeup image $x_{ref}$, textual prompts for (inversion, editing) $C_{inv}$, $\{C_{edit,s}\}_{s=1}^{N}$, degree of composition $\{\alpha_s\}_{s=1}^{N}$.
**Output:** Image with makeup transformation $\tilde{x}_0$.

  1: $z_0 = \mathcal{E}(x_0)$
  2:
  3: *1. Early-stopped DDIM inversion*
  4: **for** $t = 1$ **to** $t^*$ **do**

  5:      $z_t = \frac{\sqrt{\bar{\alpha}_t}}{\sqrt{\bar{\alpha}_{t-1}}} z_{t-1} - \sqrt{\bar{\alpha}_t}\left( \sqrt{\frac{1}{\bar{\alpha}_{t-1}} - 1} - \sqrt{\frac{1}{\bar{\alpha}_t} - 1} \right) \epsilon_\theta\big( z_{t-1}, t, c \big).$

  6: **end for**
  7: $\hat{z}_0(t^*) = \frac{1}{\sqrt{\bar{\alpha}_{t*}}}(z_{t*} - \sqrt{1 - \bar{\alpha}_{t*}}\,\epsilon_\theta(z_{t*}, t^*)).$
  8: $\hat{x}_0(t^*) = \mathcal{D}(\hat{z}_0(t^*))$
  9:
10: *2. Pixel-domain Diffusion Guidance*
11: $\hat{x}_{new} = \mathcal{T}_{RGB}\big(\mu_{src}(\hat{x}_0(t^*)), \mu_{tgt}; \alpha\big)$
12: $\tilde{z}_{t*} = \mathcal{E}(\hat{x}_{new})$
13:
14: *3. Reverse sampling with cross attention composition*
15: **for** $t = t^*$ **to** $1$ **do**
16:      $\tilde{z}_{t-1} \leftarrow \text{ReverseDDIM}\Big(\tilde{z}_t; t, \text{Composition}\big(\{\alpha_s\}_{s=1}^{N}, \{C_{edit,s}\}_{s=1}^{N}\big)\Big)$
17: **end for**
18: $\tilde{x}_0 = \mathcal{D}(\tilde{z}_0)$

---

## B  ADDITIONAL ANALYSIS

### B.1  ABLATION STUDIES

#### B.1.1  EFFECTS OF GRADATION SMOOTHING

In the generating process of eye mask, gradation smoothing is essential. Without gradation smoothing, the edges of eye masks are accentuated, resulting in an unnatural outcome. Fig. 11 demonstrates that graduation smoothing makes the edge of the eye shadow natural and realistic.

#### B.1.2  EFFECTS OF EARLY-STOPPED DDIM INVERSION

Figure 12 demonstrates that early-stopping inversion provides a valuable knob for adjusting RGB makeup transformation fidelity and naturalness. Increasing $t^*$ improves target makeup representation at the affordable cost of higher computational demands, while decreasing $t^*$ preserves subject identity and ensures accurate color representation. This approach, including adjustments with $t^*$ and other parameters like $\alpha$, offers a remarkable customization capacity unavailable in conventional frameworks.

### B.2  LORA VARIATION

As mentioned in the main paper, the Dreamshaper pre-trained model and BKG1 LoRA weights are mainly used in our experiments. Fig. 13 shows the experimental results of using other LoRA

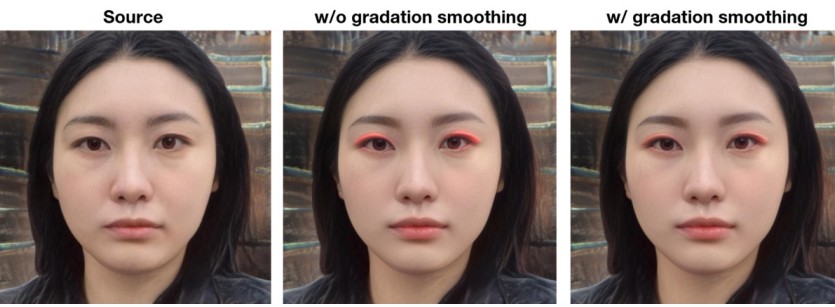

Figure 11: The effect of the gradation smoothing.

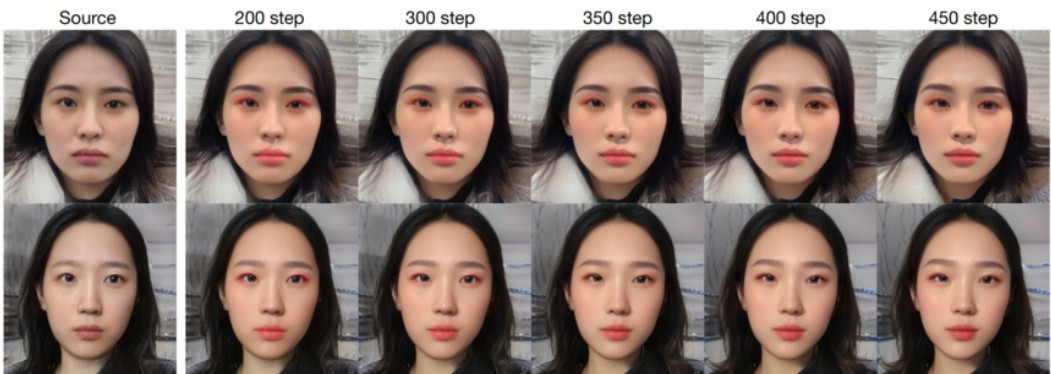

Figure 12: The effect of the early-stopped DDIM inversion step $t^*$.

weights. For comparison, asian beauty v2[4] , Korean Alike[5], Asian Cute Face[6], koreanDollLikeness v15[7], PMN 2[8] are used. The results are made with only text guidance, where the prompts are `"deep red lip"` and `"heavy eye makeup"`. The results demonstrate how diverse makeup styles can be achieved by varying LoRA weights. In this paper, we mainly leverage BKG1 LoRA which shows better identity preservation and semantic alignment.

## C  EXPERIMENTAL DETAILS

We use DDIM scheduler and set the early-stop inversion step ranging from $t^* = 200$ to $t^* = 400$. The number of reverse steps is set to 30. LoRA scale $s$ is set to 0.2. To smooth eye shadow masks, we employed a cross-shaped kernel with the size of $(12, 7)$ and performed 2 iterations of mask dilation. The textual prompts commonly used in cross attention composition are as follows:

- natural lips, natural makeup, fair skin, asian skin
- korean makeup, korean style, korean beauty, (A Classy and Cute Korean girl:1.3), cute, (Korean idol), K-pop, skm_misoo, beautiful
- 32K, high-res, (masterpiece:1.3), best quality, 8K.HDR, smooth face, 1 girl,close up face, (photorealistic:1.6), [:(detailed face:1.2):0.3]
- (Glossy lips:1.6), Gleaming lips, (fair skin:1.4), sharp focus, blusher
- (Goddess smile:1.3)
- (worst_quality:2.0) low quality, blur, deformed ugly, pixelated, cgi, illustration, cartoon, deformed, distorted, disfigured, poorly drawn

---

[4]https://civitai.com/models/76883/2731-pretty-asian-face-asian-beauty-faces
[5]https://civitai.com/models/193777/korean-alike-by-noerman
[6]https://civitai.com/models/26914?modelVersionId=32215
[7]https://civitai.com/models/26124/koreandolllikeness-v20
[8]https://civitai.com/models/106028/korean-beauty

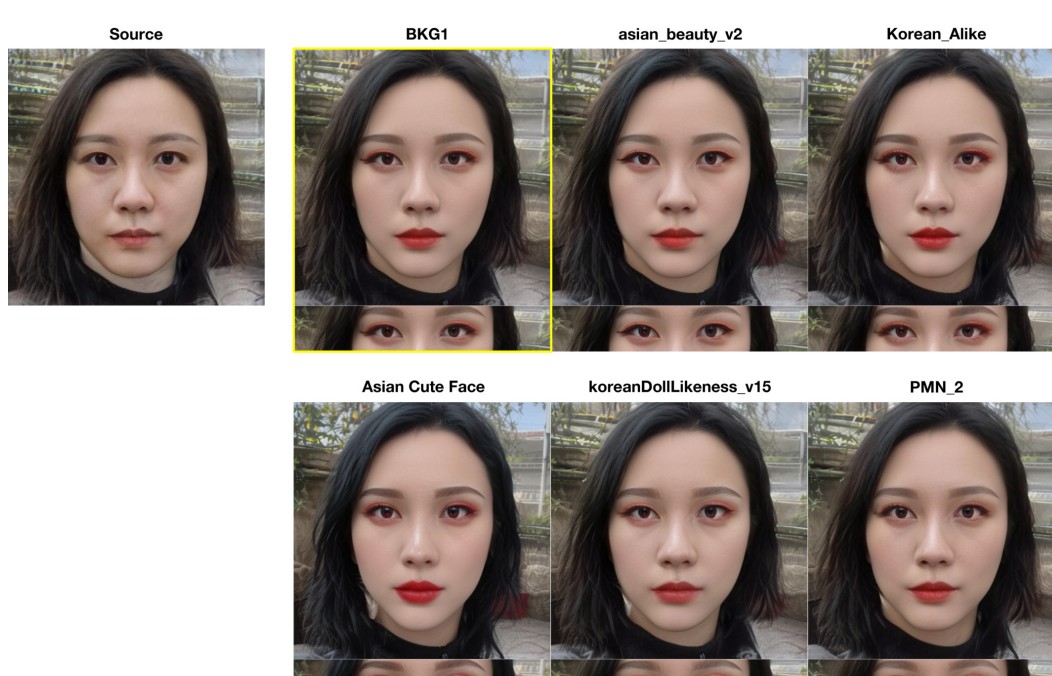

Figure 13: Results of using various LoRA weights.

The directional degree of $s$-th composition, $\alpha_s \leqslant 0$, is assigned $0.1, 0.1, 0.3, 0.7, 0.1, -0.1$ for each prompt.

## C.1 LLM

To train the language model, we constructed a QnA dataset containing information matching makeup and facial attributes. The dataset consists of $460,000$ pairs of questions and answers. Below is an example of the makeup dataset.

```
### Instruction:  Which lip colors are suitable for women with
the
following condition?  \nbronze skin, square face, angular jaw
### Response:  deep red or vivid red or dark red.
```

As a base model, we utilized dolly-v2-3b[9] and fine-tuned the model for 3 epochs using the makeup dataset. To prevent the model from forgetting language proficiency during fine-tuning, we also incorporated the natural language dataset used to train this base language model. The training objective is to generate the subsequent tokens based on the tokenized instructions in an autoregressive manner.

---

[9]Databricks, Free dolly: Introducing the world's first truly open instruction-tuned llm, https://github.com/databrickslabs/dolly, 2023.

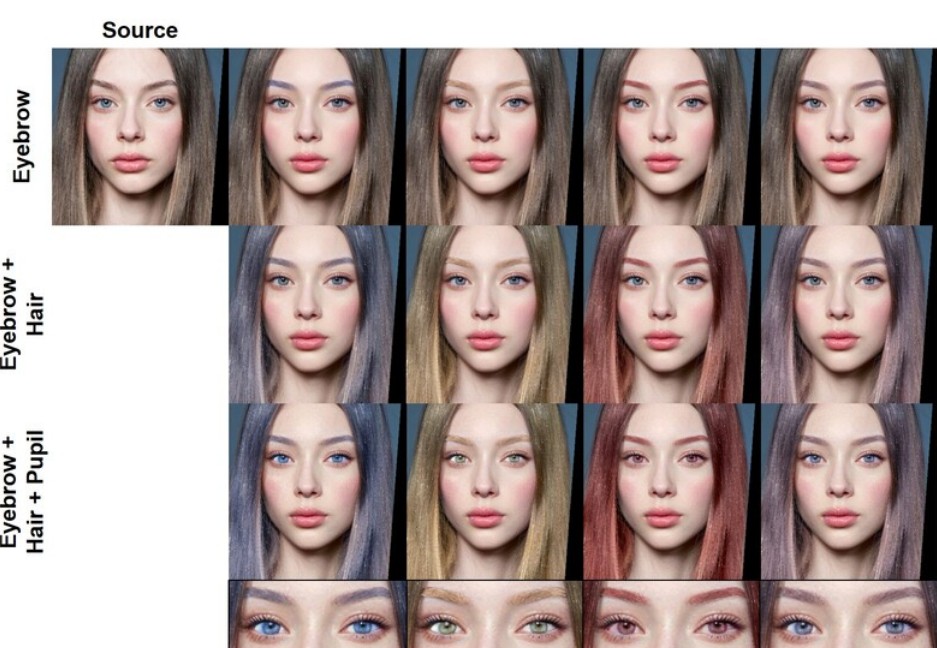

Figure 14: DreamMakeup results on RGB matching for eyebrows, pupils, and hair.

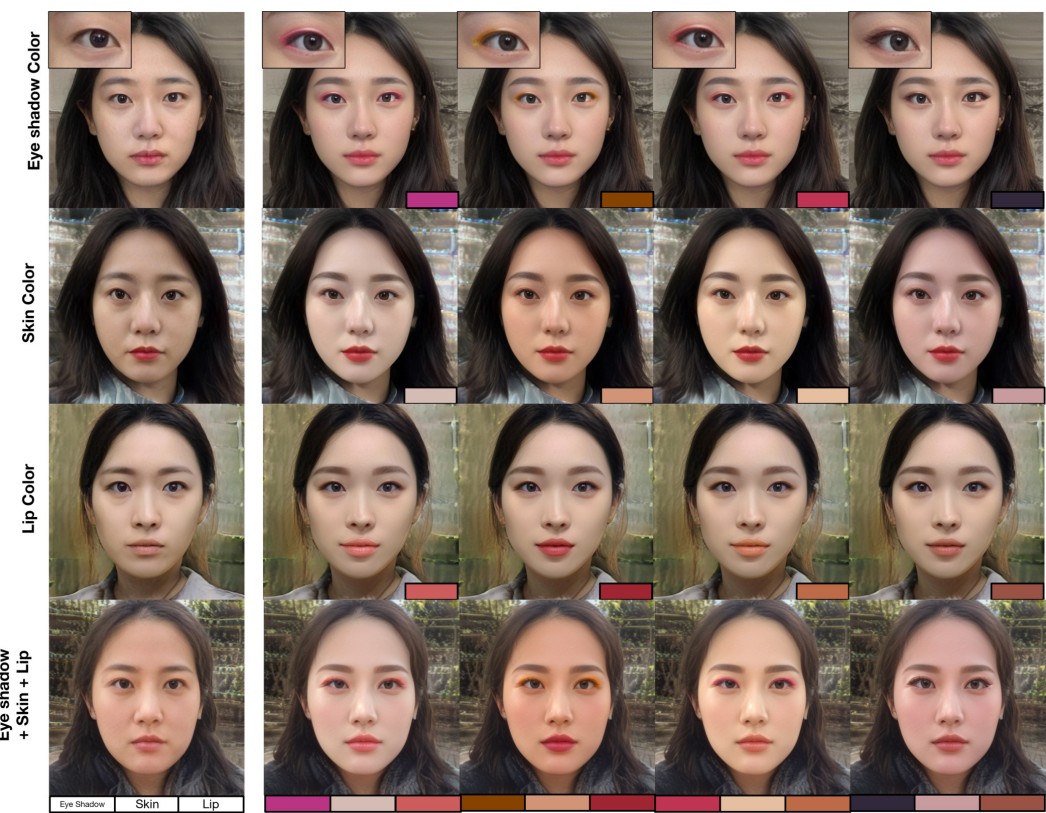

Figure 15: The skin, lip, eye shadow makeup, and their combination.

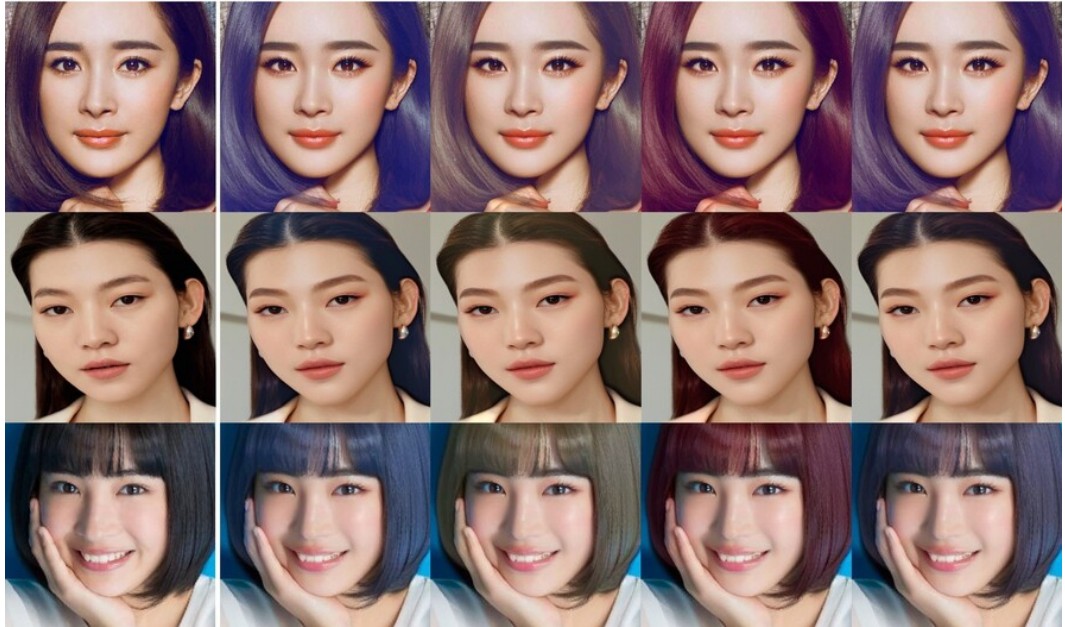

Figure 16: The results on hair coloring.

