# OpenReview forum: "DreamMakeup: Face Makeup Customization using Latent Diffusion Models"
_ICLR.cc/2025/Conference — ICLR 2025 Conference Withdrawn Submission_

### Official Review · Reviewer_42ww · 2024-11-03

**Soundness:** 2
**Presentation:** 3
**Contribution:** 2
**Rating:** 5
**Confidence:** 4

**Summary:**

This paper proposes a Diffusion-model-based generative method,, namely DreamMakeup, which can perform makeup customization with various condition inputs, including RGB colors, reference images and textual descriptions. This method employs an early-stopped DDIM inversion strategy to preserve the facial structure, then integrates the RGB colors and makeup styles in the pixel space, while injects the textual conditions by the cross-attention.

**Strengths:**

1. This paper is well-organized and easy to follow.

2. The proposed method can work well with pre-trained LoRA weights and does not require fine-tuning operations, so it seems to be computationally efficient.

3. The technical details are clearly described, which can improve the reproducibility.

4. Extensive experiments are conducted to indicate the effectiveness of DreamMakeup. And moreover, the authors also show that their method can work well with LLMs to realize a recommendation application.

**Weaknesses:**

1. For the related works, some important makeup transfer methods are not mentioned, such as SSAT [1], CSD-MT [2], and especially Stable-Makeup [3] that is also built upon the diffusion model.

2. For the reference image conditioning input, the proposed method uses the histogram matching to align the color distributions of different local facial areas. Based on my own experience, simply using such technique cannot handle some complex makeup styles well. Could the author conduct some experiments on LADN [4] or CPM [5] dataset to show the generalizationability on complex makeup styles?

3. For the experiments, the benchmark methods seem to be a bit outdated, could the authors compare their method with recently proposed GAN-based method CSD-MT [2] and Diffusion-model-based method Stable-Makeup [3]?

[1] Sun Z, Chen Y, Xiong S. Ssat: A symmetric semantic-aware transformer network for makeup transfer and removal[C]//Proceedings of the AAAI Conference on artificial intelligence. 2022, 36(2): 2325-2334.

[2] Sun Z, Xiong S, Chen Y, et al. Content-Style Decoupling for Unsupervised Makeup Transfer without Generating Pseudo Ground Truth[C]//Proceedings of the IEEE/CVF Conference on Computer Vision and Pattern Recognition. 2024: 7601-7610.

[3] Zhang Y, Wei L, Zhang Q, et al. Stable-Makeup: When Real-World Makeup Transfer Meets Diffusion Model[J]. arXiv preprint arXiv:2403.07764, 2024.

[4] Gu Q, Wang G, Chiu M T, et al. Ladn: Local adversarial disentangling network for facial makeup and de-makeup[C]//Proceedings of the IEEE/CVF International conference on computer vision. 2019: 10481-10490.

[5] Nguyen T, Tran A T, Hoai M. Lipstick ain't enough: beyond color matching for in-the-wild makeup transfer[C]//Proceedings of the IEEE/CVF Conference on computer vision and pattern recognition. 2021: 13305-13314.

**Questions:**

1. I am confused about the DDIM inversion operation, it seems to be somewhat redundant. The authors first decode the denoised estimate back to the pixel space to obtain ${\hat{x}_{0}(t^{*})}$ and then perform the makeup customization on it. My quesion is why do not directly perform it on the original image that can fully preserves the facial structure and identity information? Moreover, from Figure 12, it can be seen that different early-stopped DDIM inversion steps do not significantly affect the final generation results. This may also support to do makeup customization on the original image.

2. From Figure 5, I observe that some results of PSGAN and EleGANt have been cropped (see rows 2, 4 and 5), could the authors explain the reasons for doing such cropping operation. And does this operation lead to misalignment between the result and the source image, thus degrading the performance of PSGAN and EleGANt on the LPIPS evaluation metric?

3. I am wondering the robustness of the proposed DreamMakeup method. Could it handle reference image inputs with large head pose/facial expression variations, different skin color/illumination conditions, and occlusions in the facial regions?

---

### Official Review · Reviewer_JUK8 · 2024-11-03

**Soundness:** 3
**Presentation:** 3
**Contribution:** 2
**Rating:** 6
**Confidence:** 4

**Summary:**

The paper introduces DreamMakeup, a facial makeup customization method based on diffusion models. This approach overcomes the limitations of GAN-based methods, offering greater controllability and customization capabilities. Users can customize makeup through RGB colors, reference images, and text descriptions. DreamMakeup outperforms existing methods in makeup transfer tasks and is computationally efficient. Additionally, the paper explores the potential of combining this approach with LLMs for personalized makeup recommendations.

**Strengths:**

1. The paper utilizes diffusion models for facial makeup customization, fully leveraging the capabilities of conditional diffusion models to support comprehensive customization through text descriptions, colors, and reference images.
2. The visual effects demonstrated in the paper are impressive, allowing for makeup transformations on various facial areas, such as eyeshadow, hair, and lips.
3. The paper is well-structured, with a simple and direct methodology.

**Weaknesses:**

1. Using early stopped DDIM inversion to maintain structural consistency is not a novel contribution and theoretically might lead to some identity information loss: utilizing early stopped DDIM inversion to maintain structural consistency.
2. Some metrics are provided in Table 1, but I would like to know how much loss there is in identity similarity.
3. The use of synthetic data has somewhat alleviated the dependency on real data. However, when dealing with makeup effects for specific ethnicities or skin tones, more customized data may be required.

**Questions:**

1. Is my understanding of the article's process correct? Given an input image, DDIM inversion is performed, followed by direct prediction of $x_0$​ at time step $t^*$ and decoding through the decoder. The makeup is applied by considering color matching in the pixel domain, then re-encoded with noise through the encoder, and finally denoised back to the result.
2. In some visual results, lighting effects are mistakenly transferred as makeup. What are some better considerations to address this issue?
3. What does the specific inference process look like?

---

### Official Review · Reviewer_cnFG · 2024-11-04

**Soundness:** 2
**Presentation:** 2
**Contribution:** 1
**Rating:** 3
**Confidence:** 3

**Summary:**

The proposed method aims to customize face makeup by leveraging the powerful controllability of diffusion model. For image fidelity, images are inverted via DDIM inversion and early stopping for structural consistency. Then, makeup is explicitly performed in RGB pixel space, and further improved with the harmonization process in the latent space of LDM.

**Strengths:**

The proposed method adopts diffusion-based model and shows reliable results with real-image.

**Weaknesses:**

The proposed method makes a main contribution by pointing out that StyleGAN-based approaches are unsatisfactory and have limited customization capabilities. This makes some sense, but some questions still remain about the aforementioned contributions.

- First, there are several make-up customization papers [1-3] that leverage the superior controllability of diffusion models. Therefore, it is unconvincing that this paper is not the first proposed method in the field, and that it is not a significant improvement compared to previous models.
- Second, the title of the proposed paper seems to show that the LDM is the core network on the generation pipeline. The main purpose is to perform face makeup customization, but this process is conducted in pixel space than latent space. LDM is utilized for harmonization. Does LDM bring significant visual improvements? It is wondered how the quantitative results would change if the face makeup are performed without the last harmonization step.
- Third, there is a lack of comparison with other diffusion-based makeup models and a lack of discussion of the limitation of the proposed method. It can be intuitively expected that each component of proposed pipeline frequently exhibit limitations for face image. In particular, explicit face segmentation in Eq. (8) is highly dependent on BisNet. It is know that this is often unable to accurately segment the facial areas due to accessories, hairstyles and so forth. In this respect, there may be limitations in editing real-world image. Although the authors claim that real-images can be edited accurately, The author insist that real-world image can be edited precisely. However the samples of results looks quite clear and easy to edit.
- Lastly, it is necessary to demonstrate failure case and discuss the limitations of proposed method.

Based on above questions and concerns, it is not convincing for me. Lack of comparison with other diffusion-based makeup models and the concerns about pixel space editing largely affect the score. To summarize, there is no significant contribution, thus leading to a negative score.

[1] Toward Tiny and High-quality Facial Makeup with Data Amplify Learning, 2024

[2] MakeupDiffuse: a double image-controlled diffusion model for exquisite makeup transfer, 2024

[3] DiffAM: Diffusion-based Adversarial Makeup Transfer for Facial Privacy Protection, 2024

**Questions:**

1. Does it cause a significant degradation if the final image is edited in pixel space without latent space, that is output is generated by only Eq. (8)? Continuing with this, it is wondered that the objective and advantages of the use of LDM is clear and reasonable in the perspective of efficacy.

2. There are some makeup papers based on diffusion models, but why is there a lack of comparison with these models in experiments?

3. Is there a rationale and clear advantage to performing makeup customization in pixel space rather than latent space in diffusion models?

4. Does it works well under a variety of input conditions, such as low-light images, occluded faces, glasses, non-aligned facial image (e.g., side-view), various expressions (e.g., smiling, opened mouth)?

---

### Official Review · Reviewer_6jmQ · 2024-11-04

**Soundness:** 2
**Presentation:** 3
**Contribution:** 2
**Rating:** 5
**Confidence:** 4

**Summary:**

This paper presents an interesting approach but lacks comprehensive validation, both in terms of dataset diversity and collaborative evaluation with LLMs, which detracts from its potential contributions.

- Insufficient Experimental Validation: The paper's experimental evaluation is limited to the Makeup Transfer (MT) dataset, which restricts the breadth of its findings. Given the complex nature of makeup styles, it is essential to validate the proposed method on diverse and challenging datasets that encompass a wider range of makeup types, lighting conditions, and facial attributes. The current evaluation does not sufficiently demonstrate the model's adaptability and robustness across real-world makeup applications.

- Inadequate Assessment of LLM Collaboration: The paper discusses collaboration with Large Language Models (LLMs) for personalized makeup suggestions, but this integration is not comprehensively evaluated. A lack of clear metrics and qualitative examples makes it challenging to ascertain the effectiveness and utility of the LLM’s involvement. Without a rigorous assessment framework, the claimed benefits of LLM integration remain speculative.

- Limitations in Visual Quality: The visual makeup outcomes, as presented, lack a cohesive aesthetic across varied makeup styles. The generated makeup does not consistently achieve realistic or visually appealing results, which could be attributed to the model's reliance on latent diffusion without further refinement steps on more complex makeup datasets. The paper could benefit from comparative visual evaluations with state-of-the-art methods to underscore any improvements.

**Strengths:**

Please refer to Summary.

**Weaknesses:**

Please refer to Summary.

**Questions:**

Please refer to Summary.

---

### Note · Authors · 2024-11-15

**Comment:**

We genuinely thank the reviewers for reviewing our paper. We would like to fully address all the issues, but for this we would need a few more time. After careful consideration, we decided to withdraw our paper.

**Withdrawal Confirmation:**

I have read and agree with the venue's withdrawal policy on behalf of myself and my co-authors.